# Targeted treatment of injured nestmates with antimicrobial compounds in an ant society

Erik. T. Frank ®[1,2] ✉, Lucie Kesner ®[3], Joanito Liberti ®[1,3], Quentin Helleu ®[1,4], Adria C. LeBoeuf ®[5], Andrei Dascalu[1], Douglas B. Sponsler[2], Fumika Azuma[6], Evan P. Economo[6,7], Patrice Waridel ®[8], Philipp Engel ®[3], Thomas Schmitt ®[2] & Laurent Keller[1] ✉

Infected wounds pose a major mortality risk in animals. Injuries are common in the ant *Megaponera analis*, which raids pugnacious prey. Here we show that *M. analis* can determine when wounds are infected and treat them accordingly. By applying a variety of antimicrobial compounds and proteins secreted from the metapleural gland to infected wounds, workers reduce the mortality of infected individuals by 90%. Chemical analyses showed that wound infection is associated with specific changes in the cuticular hydrocarbon profile, thereby likely allowing nestmates to diagnose the infection state of injured individuals and apply the appropriate antimicrobial treatment. This study demonstrates that *M. analis* ant societies use antimicrobial compounds produced in the metapleural glands to treat infected wounds and reduce nestmate mortality.

Infections are a major mortality risk in animals[1,2], with the risk of transmission of contagious pathogens being particularly life-threatening in group living animals[3]. This has led to a suite of pathogen-induced changes in social interactions, like social distancing, sickness cues, and medical care[4–6]. In injured individuals, the major barrier against infections (the cuticle or epidermis) is damaged and therefore provides an easy entry point for life-threatening infections[7]. Recently, several mammals have been shown to lick wounds to apply antiseptic saliva[1,5]. However, the efficacy of these behaviors remains largely unknown and occur irrespective of the state of the wound.

In social insects, interactions to combat pathogens range from preventive measures like nest disinfection or allogrooming to moribund individuals leaving the nest to die in isolation or the destructive disinfection of their infected brood[3,8–10]. But if and how social insect colonies care for injured individuals that were exposed to pathogens remains poorly understood. Workers of the predatory ant *Megaponera analis* have been shown to care for the injuries of nestmates[2,7], which are common because this ant feeds exclusively on pugnacious termite species[11]. As many as 22% of the foragers engaging in raids attacking termites have one or two missing legs[2]. Injured workers are carried back to the nest where other workers treat their wounds, by licking and grooming the wound during the first three hours after injury[7]. When the wounds of injured workers are not treated by nestmates, 90% of the injured workers die within 24 h after injury[7], but the mechanisms behind these treatments are unknown. The aim of this study is therefore to identify the cause of death in injured individuals and the potential mechanisms involved in the detection and treatment of injuries.

[1]Department of Ecology and Evolution, Biophore, University of Lausanne, CH-1015 Lausanne, Switzerland. [2]Department of Animal Ecology and Tropical Biology, Biocenter, University of Würzburg, D-97074 Würzburg, Germany. [3]Department of Fundamental Microbiology, Biophore, University of Lausanne, CH-1015 Lausanne, Switzerland. [4]Structure et Instabilité des Génomes, Muséum National d'Histoire Naturelle, CNRS UMR 7196, INSERM U1154, 43 rue Cuvier, F-75005 Paris, France. [5]Department of Biology, University of Fribourg, CH-1700 Fribourg, Switzerland. [6]Biodiversity and Biocomplexity Unit, Okinawa Institute of Science and Technology Graduate University, Onna 904-0495, Japan. [7]Radcliffe Institute for Advanced Study, Harvard University, Cambridge 02138, USA. [8]Protein Analysis Facility, Génopode, University of Lausanne, CH-1015 Lausanne, Switzerland. ✉e-mail: erik.frank@uni-wuerzburg.de; laurentkeller01@gmail.com

Here we show that the gram-negative bacterium *Pseudomonas aeruginosa* can cause lethal infections in injured *M. analis* workers. We found that infected wounds are treated more often than sterile wounds and that this correlates with changes in the cuticular hydrocarbon profile of infected individuals. We describe the use of the metapleural gland to treat infected wounds and quantify its content, identifying 112 chemical compounds and 41 proteins in the gland's secretion, half of which have antimicrobial or wound healing properties. Overall, this study demonstrates that the targeted use of antimicrobials to treat infected wounds is effective in decreasing mortality in injured individuals.

## Results and discussion

### Pathogen identification and mortality cause

To investigate whether the high mortality of injured individuals is due to infection by pathogens, we collected soil from the natural environment and applied it to the wounds of experimentally injured workers (i.e., a sterile cut in the middle of the femur on the hind leg of an otherwise healthy ant). After 2 h, injured ants exposed to the soil (hereafter referred to as "infected ants") had ten times higher bacterial loads in the thorax than similarly injured individuals exposed to sterile phosphate-buffered saline (PBS, hereafter referred to as "sterile ants"; least square means: degrees of freedom (DF) = 35; $t = -3.08$; $P = 0.01$; Fig. 1a and Supplementary Table 1). After 11 h, bacterial load further increased in infected ants (least square means: DF = 35; $t = -3.66$; $P = 0.003$), while there was no such increase in sterile ants (least square means: DF = 35; $t = -1.54$; $P = 0.26$; Fig. 1a). As a result, after 11 h the bacterial load was 100 times higher in infected than sterile ants (least square means: DF = 35; $t = -5.21$; $P < 0.001$; Fig. 1a). A microbiome analysis further revealed major differences in bacterial species composition and abundance between the two groups of ants (ADONIS: $F_{(1,39)} = 17.45$; $R^2 = 0.31$; $P < 0.001$; Fig. 1b and Supplementary Fig. 1a, b), with three potentially pathogenic bacterial genera increasing in absolute abundance in the thorax of infected ants at both time-points: *Klebsiella*, *Pseudomonas* and *Burkholderia* (Fig. 1b and Supplementary Table 2).

To investigate if the differences in bacterial abundance and composition affect survival, we either placed infected and sterile ants in their colony or kept them in isolation. The mortality after 36 h was much lower for infected ants kept with their nestmates (22%) than for infected ants kept in isolation (90%, least square means: DF = 154; $Z = -2.759$; $P = 0.02$; Fig. 1c). By contrast, there was no significant difference between the mortality of sterile ants kept with their nestmates or in isolation (least square means: DF = 154; $Z = 1.04$; $P = 0.89$; Fig. 1c). The mortality of infected ants was also not significantly different than the mortality of sterile ants when these individuals were kept with their nestmates (least square means: DF = 154; $Z = -0.630$; $P = 1$; Fig. 1c). Overall, these data demonstrate that *M. analis* workers are capable of effectively treating wounds that have been exposed to soil pathogens through social interactions.

By culturing the soil medium on agar plates, we were able to isolate three potential pathogens (the endosymbiotic bacterium *Burkholderia* sp. and its fungal host *Rhizopus microsporus*, and the bacterium *Pseudomonas aeruginosa*, Fig. 1b and Supplementary Fig. 1c, d). While the application of *B.* sp. and *R. microsporus* (separately or together) on wounds did not significantly decrease survival (Supplementary Figs. 2b and 3), the application of *P. aeruginosa* (OD = 0.05), a bacterium widespread in various environments[12], caused a mortality of 93% within 36 h (Fig. 2a). Since the treatment with only *P. aeruginosa* showed a similar mortality as the treatment with all soil pathogens (90%, Supplementary Fig. 3), we only used *P. aeruginosa* in subsequent infection assays to better control pathogen load.

Like the experiments where the soil was applied to the wound, the presence of nestmates was also effective in decreasing the mortality of injured workers exposed to a known concentration of *P. aeruginosa* (OD = 0.05). While the mortality of infected ants kept in isolation was 93%, mortality of infected ants that had been returned to their nestmates was only 8% (least square means: DF = 142; $Z = -2.94$; $P = 0.01$; Fig. 2a, Supplementary Fig. 2c, and Supplementary Table 4). There were major differences in the increase in *Pseudomonas* load after injury between ants kept with or without their nestmates (Fig. 2b & Supplementary Table 5). The bacterial load of infected ants kept with their nestmates did not increase significantly from 2 and 11 h after injury (least square means: DF = 18; $t = -0.037$, $P = 1$; Fig. 2b). By contrast, there was a 100-fold increase in bacterial load for infected ants kept in isolation (least square means: DF = 18; $t = -4.832$, $P < 0.001$; Fig. 2b).

### Wound care behavior

To study the proximate mechanisms reducing mortality of ants infected with *P. aeruginosa* when they are returned to their nestmates, we introduced injured ants (with sterile and infected wounds) to their nestmates and filmed them for 24 h. We observed that workers treated the injury of infected ants by depositing secretions produced by the metapleural gland (MG), which is located at the back of the thorax (Fig. 3a). The MG secretions, which have antimicrobial properties[13–17], were applied in 10.5% of the wound care interactions (43 out of 411). Before applying MG secretions, the nursing ant always groomed the wound first (i.e., "licking" the wound with their mouthparts). Nursing ants then collected the secretions either from their own MGs (Supplementary Movie 1), by reaching back to the opening of the gland with their front legs to collect the secretion and then licking their front legs to accumulate it in their mouth, as described in other species[15], or from the MG of the injured ant itself, by licking directly into the gland's opening (Supplementary Movie 2). Wound care with MG secretions lasted significantly longer ($85 \pm 53$ s) than wound care without MG secretions ($53 \pm 36$ s; t-test: DF = 47.92; $t = -3.09$; $P = 0.003$; Supplementary Fig. 4).

Remarkably, workers were able to discriminate between infected and sterile ants. Wound care treatment was provided more often to infected ants upon initial introduction to the nest and again 10- and 11-hours post-introduction (hierarchical generalized additive model: $P < 0.05$; Fig. 3b). Moreover, MG secretions were deposited significantly more often on wounds of infected than sterile ants between 10 and 12 h after infection (hierarchical generalized additive model: $P < 0.05$; Fig. 3c). When treatment with MG secretions was prevented (through plugging the MG opening of all ants in the subcolony), mortality of infected ants reached 100% within 36 h compared to only 33% when workers had access to the MG secretions (least square means: DF = 47; $Z = 2.99$; $P = 0.039$; Fig. 4, Supplementary Fig. 5, and Supplementary Table 6). By contrast, the plugging of the MG opening had no significant effect on the survival of sterile ants (least square means: DF = 47; Z = 0.02; $P = 1$; Fig. 4 and Fig. S5 and Table S6).

### CHC profiles as infection cues

Because cuticular hydrocarbons (CHCs) are known to be frequently used as a source of information in ants[18], we investigated whether the CHC profile of infected ants changed over the course of the infection. Immediately after injury, infected ants did not differ from sterile ants in their CHC profile (ADONIS: $R^2 = 0.013$; $F_{(1,11)} = 0.13$; $P = 0.96$; Supplementary Tables 7 and 8). This profile changed in both types of ants during the two hours after injury (Sterile ants: ADONIS: $R^2 = 0.15$; $F_{(1,16)} = 2.74$; $P = 0.03$; Infected ants: ADONIS: $R^2 = 0.13$; $F_{(1,17)} = 2.42$; $P = 0.04$), converging towards a similar profile for both types of ants (ADONIS: $R^2 = 0.008$; $F_{(1,22)} = 0.17$; $P = 0.87$). Thereafter, the CHC profile of infected ants remained unchanged until 11 h after injury (ADONIS: $R^2 = 0.063$; $F_{(1,23)} = 1.48$; $P = 0.13$), while the CHC profile of sterile ants changed significantly (ADONIS: $R^2 = 0.14$; $F_{(1,22)} = 3.54$; $P = 0.04$), thereby becoming significantly different from the CHC profile of

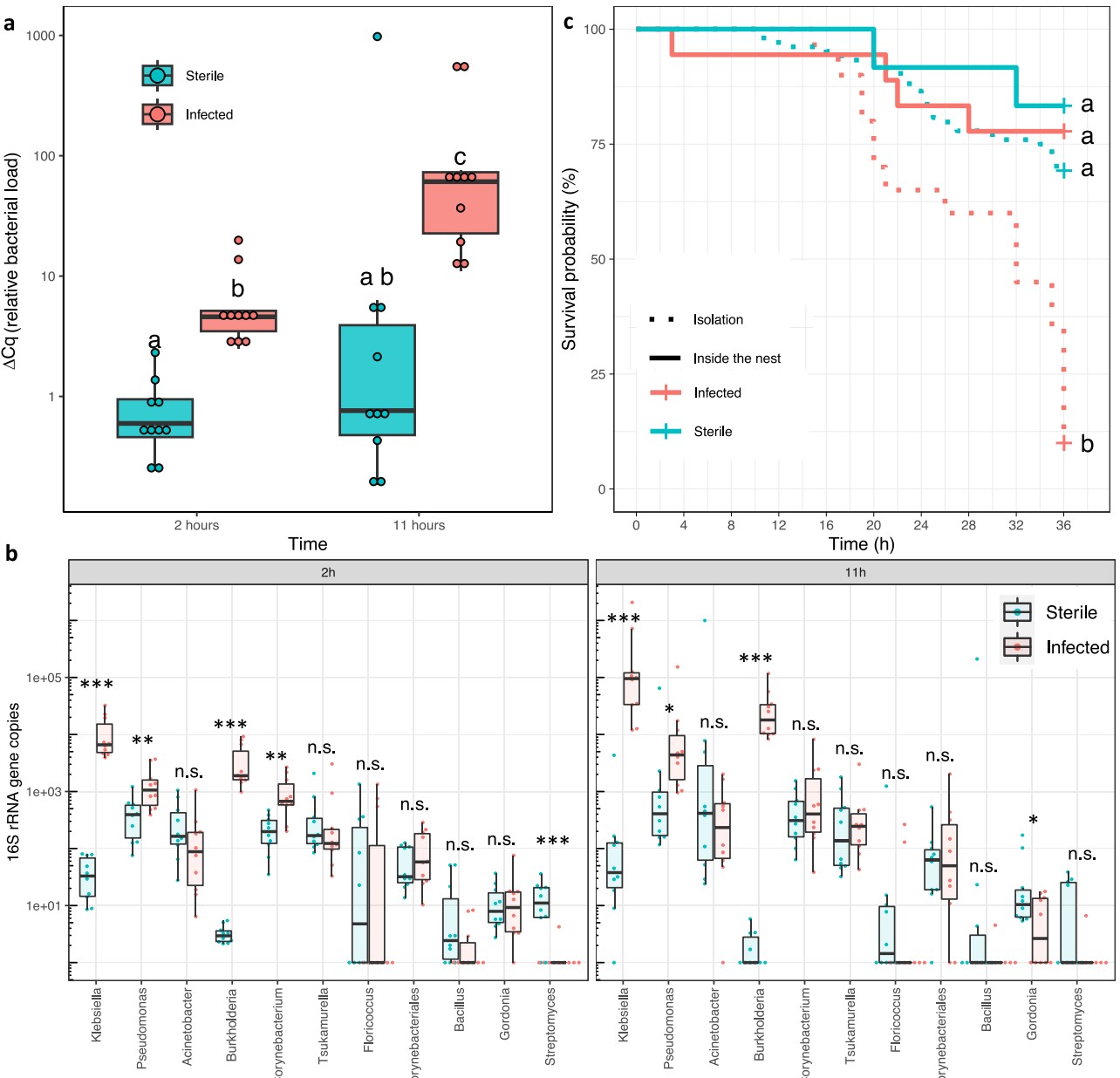

**Fig. 1 | Lethal effects and diversity of soil pathogens. a** Relative 16 S rRNA gene copies (bacterial load ΔCq) for individuals whose wounds were exposed to a sterile PBS solution (blue: Sterile), or soil pathogens diluted in PBS (red: Infected, OD = 0.1), 2 and 11 h after exposure ($n = 10$ per boxplot, see Supplementary Table 1 for statistical results). Significant differences ($P < 0.05$) are shown with different letters and were calculated using a two-sided least square means with Holm-Bonferroni correction. **b** Absolute 16 S rRNA gene copy numbers summarized at the genus-level for the 18 amplicon sequence variants (ASV) that had at least 1% relative abundance in five of the 40 analyzed ants across the experiment ($n = 10$ per boxplot). Significance is shown for pairwise comparisons between sterile (blue) and infected ants (red): ***=$P < 0.001$; **=$P < 0.01$; *=$P < 0.05$; n.s.= not significant ($P > 0.05$). Detailed

statistical results in Supplementary Table 2, significant differences were calculated with a two-sided permutation t-test with Holm-Bonferroni correction. **(c)** Kaplan – Meier cumulative survival rates of workers in isolation (dotted line) or inside the nest (solid line) whose wounds were exposed to a sterile PBS solution (blue: sterile), or soil pathogens diluted in PBS (red: infected, OD = 0.1). Significant differences ($P < 0.05$) are indicated with different letters. Detailed statistical results in Supplementary Fig. 2a and Supplementary Table 3, significant differences were calculated using a two-sided least square means with Holm-Bonferroni correction. Boxplots show median (horizontal line), interquartile range (box), distance from upper and lower quartiles times 1.5 inter-quartile range (whiskers), outliers (>1.5x upper or lower quartile). Source data are provided as a Source Data file.

infected ants (ADONIS: $R^2 = 0.19$; $F_{(1,23)} = 5.4$; $P = 0.007$). Eleven hours after injury, the CHC profile of sterile ants had converged again to the profile at 0 h (ADONIS: $R^2 = 0.03$; $F_{(1,17)} = 0.53$; $P = 0.45$), while the CHC profile of injured ants remained significantly different (ADONIS: $R^2 = 0.19$; $F_{(1,17)} = 3.82$; $P = 0.008$).

Consistent with the idea that changes in CHC profiles could provide cues on the health status of ants[9], the observed differences in the CHC profiles mostly stemmed from differences in the relative abundance of alkadienes (Supplementary Fig. 6 and Supplementary

Table 9), which are among the CHC compounds most relevant for communication in social insects[19]. These changes in the CHC profile are generally regulated by differentially expressed genes in the fat bodies[20]. To identify the genes likely responsible for the observed CHC changes between infected and sterile ants, we conducted transcriptomic analyses of the fat bodies of the same individuals. A total of 18 genes related to CHC synthesis were differentially expressed between infected and sterile ants 11 h after exposure to *P. aeruginosa* (in addition, 17 genes out of 378 that were differentially expressed were

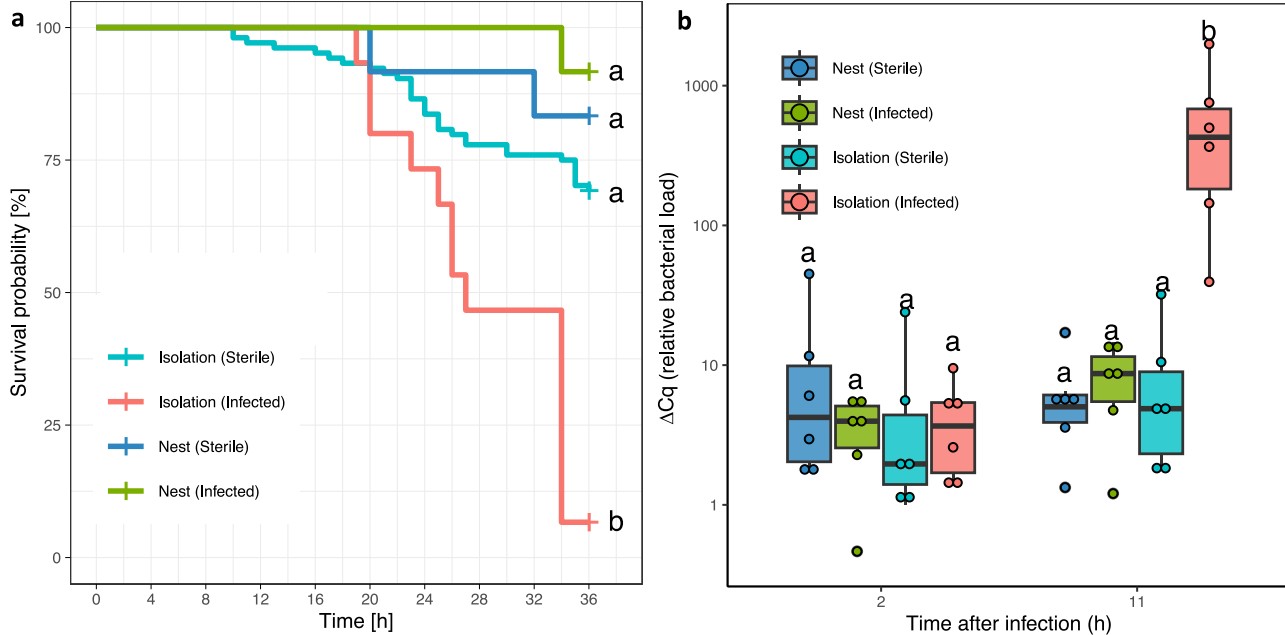

**Fig. 2 | Survival probability and pathogen load of sterile and infected ants.**
**a** Kaplan–Meier cumulative survival rates of workers in isolation or inside the nest whose wounds were exposed to *P. aeruginosa* diluted in PBS (Infected, OD = 0.05) or a sterile PBS solution (Sterile). Detailed statistical results in Supplementary Fig. 2c and Supplementary Table 3. **b** relative bacterial load (ΔCq) of *Pseudomonas* at two different time points (2 h and 11 h) for ants in isolation or inside the nest with wounds treated the same way as in Fig. 2a (Infected or Sterile). *n* = 6 per boxplot, significant differences (*P* < 0.05) are shown with different letters (Supplementary

Table 4). Boxplots show median (horizontal line), interquartile range (box), distance from upper and lower quartiles times 1.5 inter-quartile range (whiskers), outliers (>1.5x upper or lower quartile). Significant differences in both panels were calculated using a two-sided least square means with Holm-Bonferroni correction. Colors in both panels show in red: infected ants in isolation, in light blue: sterile ants in isolation, in dark blue: sterile ants inside the nest, in green: infected ants inside the nest. Source data are provided as a Source Data file.

immune genes; Fig. 5a and Supplementary Tables 10 and 11), while only two genes related to CHC synthesis were differentially expressed two hours after infection (in addition to 7 immune genes out of 164; Fig. 5b and Supplementary Tables 10 and 11).

**Antimicrobial content and efficacy of the Metapleural gland**
To quantify the capabilities of MG secretions to inhibit bacterial growth, we conducted antimicrobial assays. The growth of *P. aeruginosa* was reduced by >25% when MG secretions were included in a lysogeny broth (LB) solution compared to a control LB solution without the MG secretions (Mann-Whitney U test: *W* = 54, *P* < 0.001, Fig. 3d).

Since *P. aeruginosa* has repeatedly developed antimicrobial resistance[21] and because most antimicrobial compounds found in animal saliva are unable to inhibit the growth of *P. aeruginosa*[22], we conducted proteomic and chemical analyses of the MG secretions. The proteomic analysis revealed 41 proteins (Supplementary Fig. 7 and Supplementary Table 12), 15 of which showed molecular similarity to toxins, which often have antimicrobial properties[23]. Five proteins had orthologs known to have antimicrobial activity (e.g., lysozyme, hemocytes, 2 MRJP1-like proteins) and three with melanization, a process implicated in wound healing in insects[24,25]. There was no clear function for nine proteins including the most abundant protein (13 ± 16% of the MG's endogenous protein content), for which no ortholog could be found. The evolutionarily young gene coding this protein could be a promising candidate for antimicrobial research, with research into next-generation antibiotics often relying on antimicrobial peptides[26]. The gas-chromatography/mass-spectrometry (GC-MS) analyses of the MG further revealed 112 organic compounds (23 of which could not be identified, Supplementary Fig. 8 and Supplementary Table 13). Six of the identified compounds had antibiotic- and/or fungicide-like structures and 35 were alkaloids. While we could not identify the exact structure of the alkaloids, many of them are

known to have antimicrobial properties[27]. There were also 14 carboxylic acids, making up 52% of the secretions content (Supplementary Fig. 8 and Supplementary Table 13), probably leading to a lower pH detrimental to bacterial survival and growth[13].

The number of chemical compounds identified in the MG's secretions of *M. analis* (112) is far greater than in other ant species, where the number of compounds ranges between 1 and 35 and mostly consists of carboxylic acids rather than alkaloids or antibiotic-like compounds[13]. In other ant species, the compounds in MG secretions are generally believed to be effective against early developmental stages of pathogens, like during sterilization of the nest or as a response to recent fungal exposure[13–15]. While we also observed a prophylactic use of MG secretions directly after injury, the most intense use of the MG was at the height of the infection (10–12 h, Fig. 3c), a period where, without care, infected ants start to die (Fig. 2a). This suggests that, in addition to prophylactic substances, the MG secretions also contain antimicrobial compounds capable of combatting festering infections.

During wound care, the use of the MG secretions probably fulfils a similar role as the mammals' antiseptic saliva. This analogous role likely led to the convergent evolution of functionally similar antimicrobial and wound healing proteins[28]. However, wound care treatments are indiscriminate in mammals as they have never been observed to depend on the infected state of the wound.

This study reveals an effective behavioral adaptation to identify and treat festering infections of open wounds in a social insect. The prophylactic and therapeutic use of antimicrobial secretions to counteract infection in *M. analis* (Fig. 3c) mirrors modern medical procedures used for treating dirty wounds[29]. Remarkably, the primary pathogen in ant's wounds, *Pseudomonas aeruginosa*, is also a leading cause of infection in combat wounds, where infections can account for 45% of casualties in humans[30]. In *M. analis* the targeted treatment with antimicrobial compounds was extremely effective in preventing lethal

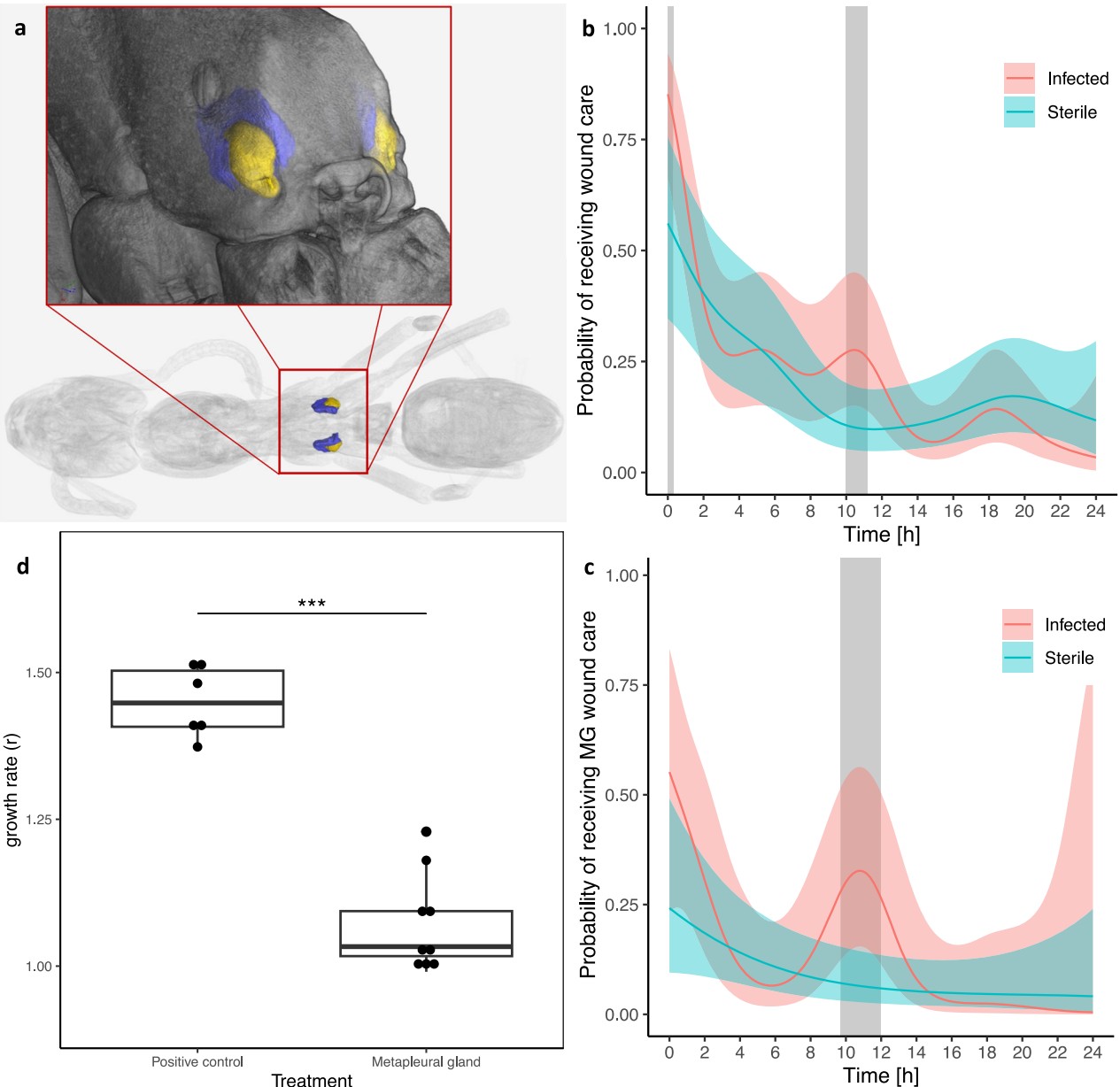

**Fig. 3 | Use and efficacy of the metapleural gland (MG) secretions during wound care. a** Micro CT scan showing the location of the MG. Blue: secretory cells; yellow: atrium. **b** Probability of receiving wound care over 24 h fitted with a hierarchical generalized (binomial) additive model (HGAM); shaded bars indicate periods during which the probability of receiving care was significantly ($P < 0.05$) higher for infected (red; $n = 6$) than sterile (blue; $n = 6$) individuals. The line represents the predicted probability by the HGAM, with the colored shaded area representing the 95% confidence interval. **c** Probability of receiving antimicrobial wound care with metapleuralgland (MG) secretions (the same ants as in Fig. 3b), modeled with identical HGAM specifications. **d** Bacterial growth assay for *P. aeruginosa* either in LB broth (positive control, $n = 6$) or LB broth with MG secretions (Metapleural gland $n = 9$). Two-sided Mann-Whitney U test: $W = 54$, $P < 0.001$. Boxplots show median (horizontal line), interquartile range (box), distance from upper and lower quartiles times 1.5 inter-quartile range (whiskers), outliers (>1.5x upper or lower quartile). Source data are provided as a Source Data file.

bacterial infections by *P. aeruginosa*. This could potentially lead to promising new medical compounds to cure infections in human societies.

## Methods

The research conducted in this study complies with all relevant ethical regulations and was approved by the park management of Office Ivoirien des Parcs et Réserves (OIPR) in Côte d'Ivoire as part of the bilateral research agreement between Germany (represented by the University of Würzburg) and Côte d'Ivoire (represented by OIPR). The ants collected for this study are part of the bilateral research

agreement under research permit number N°018 / MINEDD / OIPR / DZ.

### Experimental design

The study was conducted in a humid savannah woodland located in the Comoé National Park, northern Côte d'Ivoire (Ivory Coast), at the Comoé National Park Research Station (8°46'N, 3°47'W)[31]. Experiments, observations, and sample collections in the Comoé National Park were carried out from April to June 2018, February, April to June and September to October 2019, April 2020 and October 2022. *Megaponera analis* is found throughout sub-Saharan Africa from 25°S

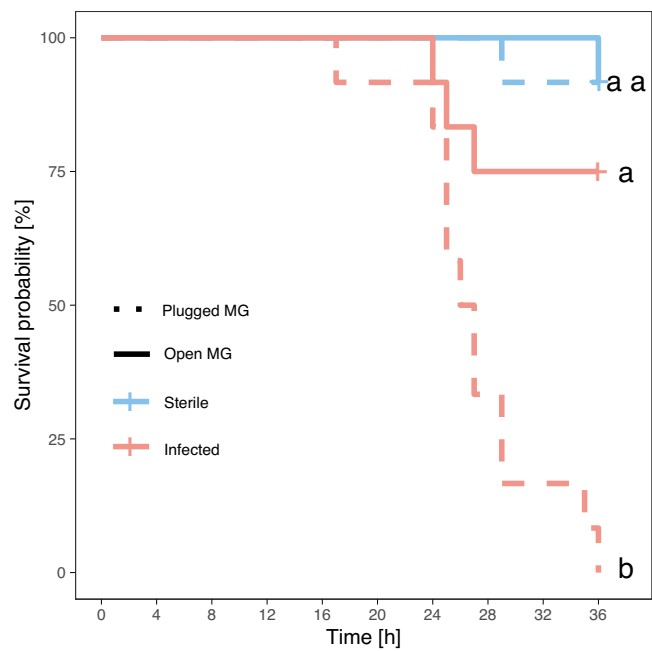

**Fig. 4 | Effect of MG secretions on survival of sterile and infected ants.**
Kaplan–Meier cumulative survival rates of workers inside sub-colonies with a
plugged metapleuralgland (MG) opening (dotted line) or with an unmanipulated
MG opening (solid line) whose wounds were exposed to a sterile PBS solution (blue,
sterile $n = 12$) or *P. aeruginosa* diluted in PBS (OD = 0.05) (red, infected $n = 12$).
Detailed statistical results in Supplementary Fig. 2d and Supplementary Table 6,
significant differences were calculated using a two-sided least square means with
Holm-Bonferroni correction. Additional data for ants in isolation can be found in
Supplementary Fig 5. Source data are provided as a Source Data file.

to 12°N[32] and known to show monophasic allometry within its worker
sizes[33]. We thus divided the workers into majors (head width > than
2.40 mm), minors (head width <1.99 mm), and intermediates (head
width 2.40 – 1.99 mm). All experiments were carried out on the minor
caste, the individuals most frequently injured[2]. All field studies were
conducted in accordance with local legislation and permission by the
Office Ivoirien des Parcs et Réserves (OIPR).

## Laboratory colonies

Eleven colonies, including queen and brood (colony size $1083 \pm 258$
ants), were excavated and placed in artificial nests in the field stations
laboratory. PVC nests (30x20x10 cm) were connected to a 1x1m
feeding arena. The ground surface was covered with soil from the
surrounding area (up to a height of 2 cm). Colonies were fed by placing
in the feeding arena *Macrotermes bellicosus* termites collected from the
surrounding area. These termites were found by scouts and triggered
raiding behavior. The laboratory windows were kept open to maintain
a natural humidity, temperature, and day-night cycle (light regime).

## Survival of injured ants

To quantify the lethality of various types of pathogens, workers were
injured by a sterile cut in the middle of the femur on the hind leg and
had the fresh wound submerged for 2 s in a 10 µL phosphate-buffered
saline (PBS) solution with a known pathogen concentration (~$10^6$
bacteria of either the mix of soil pathogens or isolated pathogen).
Afterwards the injured ants were placed inside a cylindrical glass
container with a diameter of 3 cm and a height of 5 cm. Before the
experiments, the glass containers were filled with 1 cm of surface soil
and placed for 3 h at 220 °C in an oven together with the forceps and
scissors for sterilization. Nest-like humidity was created by moistening
the soil with 1 mL of sterilized water (boiled for 10 min) and covered
with aluminum foil. The isolation experiments were conducted at 24 °C

in a sterile room. To test for mortality, the injured ant was checked
upon once per hour for the next 36 h, if no reaction was observed even
after shaking the container the ant was classified as dead. To ensure
replicability of the survival experiments a negative (sterile PBS solu-
tion) and positive control (using a sterile PBS solution with 0.1 optical
density (OD) of *Pseudomonas aeruginosa* (PSE)) were always included
during each survival experiment. The sample sizes were PBS: $n = 104$;
PSE 0.1: $n = 61$; PSE 0.05: $n = 15$ (Fig. 2a and Supplementary Fig. 3). For
all sample sizes and statistics see Supplementary Fig. 2.

To test if wound care by nestmates reduces the mortality of
infected ants, we first marked 10 ants per colony during a raid. 24 h
later, all marked ants were injured in the same way as in the isolation
trial. Afterwards, the wound of the injured ant was either exposed to
a sterile PBS solution (sterile), a sterile PBS solution containing 0.05
OD of *P. aeruginosa* or a sterile PBS solution containing 0.1 OD of
surface soil pathogens (grown on agar plates). Isolation trials with
the same treatments were always conducted in parallel. Sample size
of in nest survival experiments: PBS: $n = 12$; PSE 0.05: $n = 12$; Soil
$n = 18$ (Figs. 1 and 2a). For all sample sizes and statistics see Supple-
mentary Fig. 2.

To test the importance of the Metapleural gland (MG) secretions
we divided three colonies (~1000 ants) into two equally sized sub-
colonies including brood. The queens were left with a small retinue of
20 workers in a separate container. For each colony we plugged the
MG opening of all workers of one of the sub-colonies with acrylic color,
while for the other sub-colony acrylic color was placed on the thorax of
the workers. Afterwards the experimental procedure was identical to
the wound care experiment described above. Eight individuals were
removed from each sub-colony, marked, and wounded and the wound
exposed to either a sterile PBS solution ($n = 4$ per sub-colony) or a
sterile PBS solution containing 0.05 OD of *P. aeruginosa* ($n = 4$ per sub-
colony). Sample sizes were thus in total: Sterile: $n = 12$; Infected: $n = 12$
for both treatments (with and without plugged MG opening; Fig. 4). In
addition, to quantify any effects of the plugging of the MG opening on
individuals, a parallel experiment with $n = 12$ Sterile and $n = 12$ Infected
ants with or without plugged MG opening was run in isolation (Sup-
plementary fig. 5 and Supplementary Table 6).

The pathogen concentration was measured by optical density
(OD) using a portable Ultrospec 10 cell density meter (Biochrom) with
sterile PBS as solvent. For the soil pathogens, we collected surface soil
in the surrounding area of the nest and grew it over 36 h on agar plates
(Supplementary Fig. 1c). For *P. aeruginosa* we created cultures of the
isolated strains (Supplementary Fig. 1d) in the field lab from frozen
samples kept in Tryptic Soy Broth (TSB) medium with 25% glycerol
(stored and transported at −23 °C). After replating the bacterial culture
once on a fresh plate, we waited 16 h before applying the pathogen on
fresh wounds. For all experiments, we used trypticase soy agar (TSA)
plates to culture the bacteria.

## Treatment of wounds by nestmates

To quantify the wound care behaviors inside the nest, we filmed the
ants using a Panasonic HC-X1000 and analyzed the videos using VLC
media player v.3.0.16 Vetinari (intel 64 bit). The wound of the injured
ant was either exposed to a sterile PBS solution (Sterile) or a sterile PBS
solution containing 0.05 OD of *P. aeruginosa* (Infected).

All manipulated ants were placed in front of the nest entrance
directly after a raid and the nest was filmed for the subsequent 24 h.
Only one trial was conducted per colony with a total of four injured
ants per colony (two sterile and two infected), in a total of three
colonies (i.e., $n = 6$ infected ants and $n = 6$ sterile ants, Fig. 3b, c). The
observed wound care behaviors were classified into two categories:
(1) wound care: a nestmate cleans the open wound with its mouthparts;
(2) metapleural gland (MG) care: a nestmate collects MG secretions
either from its own gland (Supplementary Movie 1) or from the injured
ant (Supplementary Movie 2) in its mouth before caring for the wound.

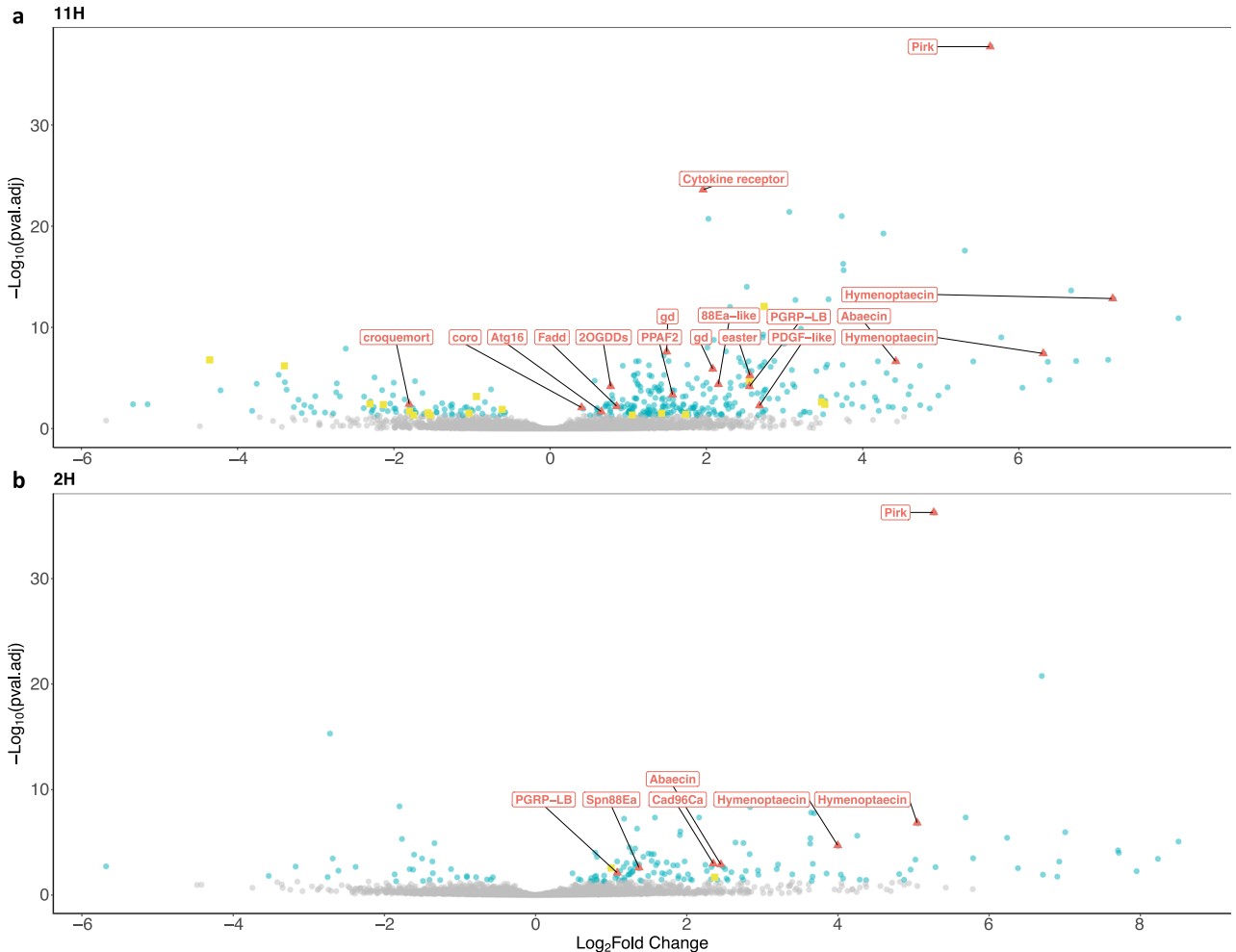

**Fig. 5 | Differential gene expression between sterile and infected ants 2 and 11 h after injury.** Infection triggers changes in the expression of hundreds of genes. Volcano plot illustrates the fold up- and down-regulation of immune-related genes (red triangles, Supplementary Table 11) and lipids and CHC-related genes (yellow squares, Supplementary Table 10), 11 h (**a**) and 2 h (**b**) after infection. Positive Log2FoldChange values correspond to genes up-regulated in infected ants when compared to sterile ants, while negative values are down-regulated in infected ants. Significant differences were calculated using a two-sided Wald test and corrected for multiple testing using the Benjamini and Hochberg method (genes with significant differential expression are marked in blue, i.e. adjusted *P*-value < 0.05). Source data are provided as a Source Data file.

These behaviors were quantified for the first 24 h and summarized in 10 min intervals.

### Sample collection protocol for chemical, genetic, and microbial analyses

To quantify the cuticular hydrocarbon (CHC) profile (Supplementary Fig. 6), differential gene expression (Fig. 5), and pathogen load in the thorax content (Figs. 1a and 2b), we used the same experimental design as used to quantify the survival of injured ants (see above). In total, 30 sterile and 30 infected ants (*P. aeruginosa* OD = 0.05) were prepared, 12 sterile and 12 infected ants were kept in isolation, another 12 sterile and infected ants were placed inside the nest and 6 sterile and 6 infected ants were collected immediately after injury. Sterile or infected ants were then collected from their enclosures at either 2 or 11 h after manipulation (*n* = 6 per treatment). Only ants that did not die until these time points were used for further analyses. The collected ants were then first placed in hexane for 10 min to extract the CHC profile. The gaster was then cut off and placed in RNAlater for genetic analyses and the thorax was placed in 100% ethanol for the microbial analyses. The samples for genetic and microbial analyses were brought to the University of Lausanne and the samples for chemical analyses to the University of Würzburg. Another 10 sterile and 10 infected ants

kept in isolation were further observed for a total of 36 h to ensure that the survival curves in this experiment resembled those in Fig. 2a.

### Pathogen identification & isolation

To isolate potential pathogens, we collected surface soil in the surrounding area of the nest and grew its water extract on Sarborough Dextrose Agar (SDA) plates for 36 h at local temperature to get the 'soil pathogen mix' (Supplementary Fig. 1c). Most of the plate was at first overgrown by black mycelia identified as *Rhizopus microsporus*. This fungus is known to contain symbiotic bacteria *Burkholderia* sp. (species not identified), as we confirmed by Sanger sequencing and bacterial community analysis (Fig. 1c). After several days, the culture started to show several large colonies of slimy microorganisms that were isolated and identified as *Pseudomonas aeruginosa* (Supplementary Fig. 1d). For infection assays (Supplementary Fig. 3), we isolated *Burkholderia* and its fungal host *Rhizopus* from each other by repeated passaging with antifungal nystatin (0.25 μl/ml in M9 agar, 30 °C) or antibacterial ciprofloxamin (0.02 mg/ml in SDA). No representatives of *Klebsiella* were isolated.

The identification of all microorganisms was done by preparation of a PCR-ready DNA from a piece of biomass as described in Lõoke et al.[34]. PCR reactions were done with universal fungal primers ITS5 5′-T

CCTCCGCTTATTGATATGC-3′ and ITS4 5′-GGAAGTAAAAGTCGTAAC AAGG-3′[35] (GoTaq polymerase, Tm = 55 °C, elongation for 40 s) and commonly used universal bacterial primers 27F 5′-AGRGTTYGAT YMTGGCTCAG-3′ and 1492 R 5′-GGTTACCTTGTTACGACTT-3′ (same protocol but elongation for 1 min 20 s). The presence of PCR amplicons was verified on electrophoresis gel and the fragments were sent for Sanger sequencing. The resulting sequences were then screened against NCBI database with nucleotide BLAST[36].

## Microbiome analysis and bacterial load quantification

To quantify the composition of the bacterial community in workers, we used the Powersoil DNA isolation kit (MO Bio) to extract DNA from *M. analis* thoraxes kept in RNAlater. Samples were homogenized by vortexing for 10 min, followed by two times 45 sec bead beating at 6 m/s using a FastPrep24™ 5 G homogenizer. Then we continued according to the kit's manual and eluted DNA in 100 µL of nuclease-free water.

Bacterial loads were quantified with a QuantStudio5 qPCR instrument (Applied Biosystems) using the reaction set-up described in Kešnerová et al.[37] and the thermal cycling conditions recommended for SYBR® Select Master Mix. For quantifying total bacterial loads, we used our designed primers #1047 5′-AGGATTAGAT ACCCTRGTAGTC-3′ and #1049 5′-CATSMTCCACCRCTTGTGC-3′ (at doubled 0.4 µM concentration). For specific targeting we used *P. aeruginosa* primers #1209 5′-GTAGATATAGGAAGGAACACCAG-3′ and #1210 5′-GGTATCTAATCCTGTTTGCTCC-3′ and for normalization to the host's housekeeping gene we used *M. analis* 28S rRNA gene primers #1207 5′-CTGCCCGGCGGTACTCG-3′ and #1208 5′-A CCGGGGACGGCGCTAG-3′. Serial dilutions (10x) showed that these primers performed with an amplification efficiency (E) of 1.86 ($R^2 = 0.99$), 2.00 ($R^2 = 0.99$), and 1.87 ($R^2 = 0.99$), respectively.

Total bacterial (Fig. 1a) and *P. aeruginosa* (Fig. 2b) 16 S rRNA gene (target) copy numbers were expressed relatively to *M. analis* 28 S rRNA gene copy numbers (host) based on the following equation: $\Delta Cq = 2^{\wedge}(Cq_{host} - Cq_{target})$, where Cq was the measured 'quantification cycle' value. To calculate the total 16 S rRNA gene copy number in 1 µl of each DNA sample that was used for absolute bacterial abundance based on ASV counts (Fig. 1b) we used the equation $n = E^{\wedge}(intercept - Cq)$[38], where standard curve's *intercept* = 38.23[37].

## 16 S rRNA gene amplicon-sequencing

16 S rRNA gene amplicon sequencing data were obtained from 40 experimental samples, a mock sample (to verify consistency of the MiSeq run compared to previous studies in our group), two blank DNA extractions and a negative PCR control with only $H_2O$. We followed the Illumina 16 S metagenomic sequencing preparation guide (https://support.illumina.com/documents/documentation/chemistry_documentation/16s/16s-metagenomic-library-prep-guide-15044223-b.pdf) to amplify and sequence the V4 region of the 16 S rRNA gene. Primers for the first PCR step were 515F-Nex (TCGTCGGCAGCGTCAG ATGTGTATAAGAGACAGGTGCCAGCMGCCGCGGTAA) and 806R-Nex (GTCTCGTGGGCTCGGAGATGTGTATAAGAGACAGGGACTACHVGGG TWTCTAAT). PCR amplifications were performed in a mix of 12.5 µL of Invitrogen Platinum SuperFi DNA Polymerase Master Mix, 5 µL of MilliQ water, 2.5 µL of each primer (5 µM), and 2.5 µL of template DNA. PCR conditions were 98 °C for 30 s, 25 cycles of 98 °C for 10 s, 55 °C for 20 s, and 72 °C for 20 s, and a final extension step at 72 °C for 5 min. Amplifications were confirmed by 2% agarose gel electrophoresis. The PCR products were then purified with Clean NGS purification beads (CleanNA) in a 1:0.8 ratio of PCR product to beads, and eluted in 27.5 µL of 10 mM Tris, pH 8.5. We then performed a second PCR step in which unique dual-index combinations were appended to the amplicons using the Nextera XT index kit (Illumina). Second-step PCRs were performed in a 25 µL, using 2.5 µL of the PCR products, 12.5 µL of Invitrogen Platinum SuperFi DNA Polymerase Master Mix, 5 µL of MilliQ water, and 2.5 µL of each of the Nextera XT indexing primers 1

and 2. PCR conditions were 95 °C for 3 min followed by eight cycles of 30 s at 95 °C, 30 s at 55 °C, and 30 s at 72 °C, and a final extension step at 72 °C for 5 min. The libraries were again purified using Clean NGS purification beads in a 1:1.12 ratio of PCR product to beads, and eluted in 27.5 µL of 10 mM Tris, pH 8.5. The amplicon concentrations, including the negative PCR control, mock, and blanks, were quantified by PicoGreen and pooled in equimolar concentrations (except for the water and blank samples that were kept at lower concentrations). We verified that the final pool was of the right size using a Fragment Analyzer (Advanced Analytical). MiSeq (Illumina) sequencing was then performed at the Genomic Technology Facility of the University of Lausanne, producing (2 × 250 bp) reads. We obtained a total of 979′126 paired-end reads across the 40 experimental samples.

## Analyses of 16 rRNA gene amplicon-sequencing data

Raw sequencing data (deposited at the Sequence Read Archive (SRA) under PRJNA826317) were analyzed with the Divisive Amplicon Denoising Algorithm 2 (DADA2) package v.1.20.0 in R. All functions were run using the recommended parameters (https://benjjneb.github.io/dada2/tutorial.html) except for the filtering step in which we truncated the forward and reverse reads after 232 and 231 bp, respectively. At the learnErrors step, we then set randomize=TRUE and nbases=3e8. Amplicon-sequence variants (ASVs) were classified with the SILVA database (version 138). Unclassified ASVs and any ASV classified as chloroplast, mitochondria or Eukaryota were removed with the "phyloseq" package version 1.36.0, using the "subset taxa" function. We then used the "prevalence" method in the R package "decontam" v.1.12.0 to identify and remove contaminants introduced during laboratory procedures, using the negative PCR control and the blank samples as reference. This procedure filtered out three contaminant ASVs. Four additional ASVs were removed as they clearly represented (low abundance) contaminants due to index swapping from samples of a different project that were sequenced in the same sequencing run. We then calculated absolute bacterial abundances of each ASV by multiplying the proportion of each ASV by the total 16S rRNA gene copy number of each sample as measured by qPCR. To assess differences in community structure between treatments we ran ADONIS tests after calculating Bray-Curtis dissimilarities with the absolute ASV abundance matrix and plotted ordinations based on these Bray-Curtis dissimilarities (Supplementary Fig. 1a).

To test for differences in absolute abundance of individual bacterial genera between sterile and infected ants (Supplementary Fig. 1b), we used a permutation approach (permutation t-test) as done in Kešnerová et al.[39]. To do this, we selected the ASVs that had at least 1% relative abundance across five samples (18 ASVs belonging to 11 genera). We then calculated copy numbers at the genus-level for each of the 40 samples. At each time-point (2 h and 11 h), we randomized the values of the calculated copy numbers for each genus 10,000 times and computed the *t* values for the tested effect for each randomized dataset. The *P* values corresponding to the effects were calculated as the proportion of 10,000 *t* values that were equal or higher than the observed one.

## Chemical analysis of cuticular hydrocarbons

To quantify differences in CHC profiles between infected and sterile ant workers, cuticular hydrocarbon extracts were evaporated to a volume of approximately 100 µL and 1 µL was analyzed by using a 6890 gas chromatograph (GC) coupled to a 5975 mass selective detector (MS) by Agilent Technologies (Waldbronn, Germany). The GC was equipped with a DB-5 capillary column (0.25 mm ID × 30 m; film thickness 0.25 µm, J & W Scientific, Folsom, Ca, USA). Helium was used as a carrier gas with a constant flow of 1 mL/min. A temperature program from 60 °C to 300 °C with 5 °C/min and finally 10 min at 300 °C was employed. Mass spectra were recorded in the EI mode with an ionization voltage of 70 eV and a source temperature of 230 °C. The

software ChemStation v. F.01.03.2357 (Agilent Technologies, Wald-bronn, Germany) for windows was used for data acquisition. Identification of the components was accomplished by comparison of library data (NIST 17) with mass spectral data of commercially purchased standards and diagnostic ions.

To compare the relative abundances of the different compound groups (Supplementary Fig. 6), all compounds were identified (Supplementary Table 8) and grouped either into Alkanes, Alkenes, Alkadienes or Methyl-branched alkanes for each individual.

### Antimicrobial assay

To assess the antimicrobial efficacy of MG extracts (Fig. 3d), we quantified the increase over time of the OD in a 96-well plate box, with the outer row filled with 70 µL of PBS. In total we did 3 replicates per sample with a total of 70 µL per sample. Negative control: 70 µL Luria-Bertani (LB) broth ($n = 6$). Positive control: 68 µL LB broth + 2 µL *P. aeruginosa* ($n = 6$). MG sample: 66 µL LB broth + 2 µL *P. aeruginosa* + 2 µL MG sample ($n = 9$).

For the preparation of the pathogen sample, we plated *P. aeruginosa* from a frozen stock on TSA plates. After 24 h the bacterial culture was replated. After 12 h aliquots were created using the new bacterial culture in a flask containing 15 mL of freshly sterilized LB broth for an initial OD reading. Afterwards the flask was placed in an incubator shaker (Amerex Steadyshake 757) at 180 rpm, 30 °C for the bacteria to grow. Once the OD readings were between 02–0.5 OD (the exponential growth phase) the flask was put on ice until the experiments started.

For the preparation of the MG sample, we pooled 10 MGs in 1.5 mL Eppendorf-Cups. We then froze the samples in liquid nitrogen and crushed the sample material using sterile pellets. We then added 50 µL of PBS-Buffer, vortexed shortly before centrifuging the sample for 5 min at 3000 x g at 4 °C. We then extracted 30 µL of the supernatant into a new cup and repeated the centrifugation process. Afterwards 20 µL of the supernatant was placed again in a new cup and stored at −20 °C. Preparation of the wells was conducted on ice at 4 °C to prevent bacterial growth and degradation of the MG samples. The antimicrobial assays were done in a microplate reader (Synergy H1 BioTek) at 30 °C with a 600 nm wavelength for 8 h with a double orbital shaking step after each OD reading cycle (every 10 min) at the University of Lausanne.

To calculate the intrinsic growth rate of the microbial population (r) in Fig. 3d we used the package growthcurver (v. 0.3.1) with the statistical software R v4.1.0. "r" represents the growth rate that would occur if there were no restrictions imposed on total population size.

### Sample collection for metapleural gland extracts

Due to the difficulty of collecting adequate amounts of metapleural gland secretions from the gland's atrium (it is a very sticky substance that adheres to the cuticle), we decided to remove the atrium of the gland together with the secretory cells entirely, using a microscalpel and microscissors. To avoid using a solvent we used a Thermodesorber unit coupled to a GC-MS (TD-GC-MS). For this we transported frozen ants to the University of Würzburg and did the extractions directly in the laboratory next to the TD-GC-MS. One metapleural gland from each of six worker was pooled per sample. As a control we further collected pieces of cuticle of similar size from the side of the thorax to identify any potential contaminations which might have occurred during dissections. In total 3 samples (of 6 individuals each) were analyzed (Supplementary Fig. 8, Supplementary Table 13).

### Chemical analysis of the metapleural gland

The MG samples were placed in a glass-wool-packed thermodesorption tube and placed in the thermodesorber unit (TDU; TD100-xr, Markes, Offenbach am Main, Germany). The thermodesorption tube was heated up to 260 °C for 10 min. The desorbed components were transferred to the cold trap (5 °C) to focus the analytes using $N_2$ flow in splitless mode. The cold trap was rapidly heated up to 310 °C at a rate of 60 °C per minute, held for 5 min and connected to the GC-MS (Agilent 7890B GC and 5977 MS, Agilent Technologies, Palo Alto, USA) via a heated transfer line (300 °C). The GC was equipped with an HP-5MS UI capillary column (0.25 mm ID × 30 m; film thickness 0.25 µm, J & W Scientific, Folsom, Ca, USA). Helium was the carrier gas using 1.2874 ml/min flow. The initial GC oven temperature was 40 °C for 1 min, then raised at a rate of 5 °C per min until reaching 300 °C, where it was held for 3 min. The transfer line temperature between GC and MS was 300 °C. The mass spectrometer was operated in electron impact (EI) ionization mode, scanning m/z from 40 to 650, at 2.4 scans per second. Chemical compounds were identified using the same protocol as for the CHCs.

### Proteomic analysis and sample preparation of the metapleural gland

To characterize the proteins secreted by the metapleural gland, we analyzed the proteome of the atrium of the gland, the metapleural gland secretory cells and the hemolymph (Supplementary Fig. 7). To be considered a metapleural gland protein that could mediate antimicrobial activity, proteins had to be found in the gland's atrium, and must have a higher abundance in the gland's atrium than in the hemolymph. The samples for proteomic analysis were collected in April 2020 from workers of field colonies collected in the Comoé National Park. Three types of samples were collected: Secretory: dissected secretory cells without the atrium, ($n = 6$ samples, each pooled from five dissections); Hemolymph: hemolymph was collected from the thorax by a glass microcapillary through a small wound ($n = 6$ samples, each pooled over five individuals, yielding 4–5 µL); Atrium: due to the difficulty of collecting pure MG content we chose to first widen the opening of the atrium and add 1 µl of PBS before extracting the content together with the PBS ($n = 6$ samples, each pooled from five individuals). The samples were kept at −23 °C in Lo-bind Eppendorf tubes with 5 µl of PBS together with a 1x Protease inhibitor Cocktail (Sigmafast) during transportation. Once in Lausanne, Switzerland, the samples were kept at −80 °C and processed swiftly.

Proteins were digested according to a modified version of the iST protocol[40]. Samples were resuspended in 20 µg of modified iST buffer (1% sodium deoxycholate, 10 mM DTT, 100 mM Tris pH 8.6) and heated at 95 °C for 5 min. They were then diluted with 24 µL of 4 mM $MgCl_2$ and 1:100 of benzonase in $H_2O$ and incubated 15 min at ambient temperature. 14 µL of 160 mM chloroacetamide (in 10 mM Tris pH 8.6) were then added and cysteines were alkylated for 45 min at 25 °C in the dark. After addition of 0.5 M EDTA (3 mM final concentration), samples were digested with 0.1 µg of trypsin/Lys-C mix (Promega) at 37 °C for 1 h, followed by a second enzyme addition (0.1 µg trypsin/LysC) and 1 h incubation. Two volumes of isopropanol + 1% trifluoroacetic acid (TFA) were added to one volume of sample and loaded onto an equilibrated OASIS MCX uElution plate (Waters) prefilled with SCX0 buffer (20% MeCN, 0.5% formic acid) and centrifuged. The columns were washed three times with 200 µL isopropanol + 1% TFA and once with 200 µL HPLC solvent A (2% MeCN, 0.1% formic acid). The peptide mixture was then sequentially eluted with 150 µL SCX125 buffer (20% MeCN, 0.5% formic acid, 125 mM ammonium acetate), 150 µL SCX500 buffer (20% MeCN, 0.5% formic acid, 500 mM ammonium acetate), and lastly with 150 µL basic elution buffer (80% MeCN, 19% water, 1% NH3).

Tryptic peptides fractions were dried and resuspended in 0.05% trifluoroacetic acid, 2% (v/v) acetonitrile, for mass spectrometry analyses. Tryptic peptide mixtures were injected on an Ultimate RSLC 3000 nanoHPLC system (Dionex, Sunnyvale, CA, USA) interfaced to an Orbitrap Fusion Tribrid mass spectrometer (Thermo Scientific, Bremen, Germany). Peptides were loaded onto a trapping microcolumn Acclaim PepMap100 C18 (20 mm × 100 µm ID, 5 µm, 100 Å, Thermo

Scientific) before separation on a reversed-phase custom-packed nanocolumn (75 µm ID × 40 cm, 1.8 µm particles, Reprosil Pur, Dr. Maisch). A flow rate of 0.25 µL/min was used with a gradient from 4 to 76% acetonitrile in 0.1% formic acid (total time: 65 min). Full survey scans were performed at a 120'000 resolution, and a top-speed precursor selection strategy was applied to maximize acquisition of peptide tandem MS spectra with a maximum cycle time of 0.6 s. HCD fragmentation mode was used at a normalized collision energy of 32%, with a precursor isolation window of 1.6 m/z, and MS/MS spectra were acquired in the ion trap. Peptides selected for MS/MS were excluded from further fragmentation during 60 s.

Tandem MS data were processed by the MaxQuant software (version 1.6.14.0)[41] using the Andromeda search engine[42] matching to a custom-made protein database containing 19'618 sequences of *M. analis* (July 2019 version, see section "Genome annotation" for details), supplemented with sequences of common contaminants. Trypsin (cleavage at K,R) was used as the enzyme definition, allowing 2 missed cleavages. Mass tolerance was 4.5 ppm on precursors (after recalibration) and 0.5 Da on MS/MS fragments. Only peptides with a minimal length of 7 were considered for protein identifications. Carbamidomethylation of cysteine was specified as a fixed modification. N-terminal acetylation of protein and oxidation of methionine were specified as variable modifications. All identifications were filtered at 1% FDR at both the peptide and protein levels with default MaxQuant parameters. For protein quantitation the iBAQ values[43] were used. MaxQuant data were further processed with Perseus software (version 1.6.14.0)[44] for filtering, removing contaminants and proteins identified with only 1 peptide, log2-transformation, normalization of values, and ortholog annotations.

To determine which proteins are secreted by the MG, we proceeded through a series of filters. Proteins not found in the atrium samples were eliminated and those found only in the MG atrium were considered hits. The log2 fold change between the atrium and the hemolymph were determined for all remaining proteins. Only proteins that were on average 1.5-fold more abundant in atrium samples than hemolymph were retained as hits. Percent of total iBAQ per sample was assigned to each protein to indicate abundance. Visualization of heatmaps was performed in Matlab 2020b (clustergram function). To determine whether our selected proteins were toxin-like, we ran clantox[45] (http://www.clantox.cs.huji.ac.il/). Annotations (function, orthology depth, and implications for wound healing) found in Supplementary Table 12 were determined using protein BLAST with the experimental clustered nr database[46].

The mass spectrometry proteomics data are deposited at the ProteomeXchange Consortium via the PRIDE partner repository with the dataset identifier PXD033003.

## Genome sequencing and assembly

The genome of *M. analis* was sequenced and assembled by The Global Ant Genomics Alliance (GAGA, antgenomics.dk)[47]. To identify repeats in the genome assembly, we used the package RepeatModeler v2.0, which combines three *de-novo* repeat finding programs (RECON, RepeatScout and LtrHarvest/Ltr_retriever)[48]. RepeatModeler first probes chunks of the genome assembly to find repeats, then clusters and classifies identified repeats, producing a high-quality library of consensus sequences of repeated sequence families. The consensus sequences were used with RepeatMasker to softmask (in lower-case) all repeats in the genome assembly.

## Genome annotation

To annotate the *M. analis* genome assembly we used the ab-initio gene predictors Augustus v3.3.3[49] and genemark[50] within the BRAKER2 pipeline v2.1.4[51,52]. Both tools were trained using RNA-seq reads (see below) aligned to the softmasked version of the genome with STAR v2.7[53] and protein evidence from NCBI refseq predictions for

*Acromyrmex echinatior, Atta colombica, Camponotus floridanus, Dinoponera quadriceps, Linepithema humile, Monomorium pharaonis, Ooceraea biroi, Pogonomyrmex barbatus, Solenopsis invicta, Temnothorax curvispinosus, Vollenhovia emeryi* and *Wasmannia auropunctata*. To enable functional interpretation of experimental data, we annotated gene products. First, we identified gene orthologs in *Drosophila melanogaster* to leverage the excellent quality of the functional annotation in this species. To identify the orthologs, we applied a reciprocal best blast approach using Orthologr[54] between the longest protein isoform for each gene from both species[55]. Additionally, we identified orthologs with *Apis mellifera* and *Camponotus floridanus* NCBI RefSeq gene annotations. Finally, we ran InterProScan (version 5.30–69.0 and panther-data-14.1) on all proteins using default settings. To assign functional information to *M. analis* annotated genes, we primarily used the gene ontology of *D. melanogaster* orthologs, but also results from sequence homology search on Interproscan and the Uniprot database.

To study the effect of infections on gene expression (Fig. 5), we conducted a transcriptomic analysis. We dissected the gaster from the same infected (soil OD = 0.1, $n = 20$) and sterile ants ($n = 20$), whose thorax were used for the microbiome analyses in Fig. 1a, b. Samples were collected 2 and 11 h after manipulation ($n = 10$ per timepoint) in the Comoé field research station and had the sting together with the venom reservoir removed. The gaster was dissected without solvent and stored in a RNAlater stabilisation solution at −23 °C (ThermoFisher Scientific) for later analyses at the University of Lausanne. Afterwards, the samples were homogenized with ceramic beads in 1 ml of Trizol reagent. Homogenized samples were incubated for 5 min at room temperature (RT) in Trizol, before adding Chloroform (200 µL). Samples were incubated for 5 min at RT then centrifuged (25 s at 12,000 rpm and 4 °C) and the upper aqueous layer (~500 µL) transferred to a new tube. We added Isopropanol (650 µL) and Glycogen blue (1 µL; RNAse-free, Invitrogen, 15 mg/mL, #AM9516) then vortexed and incubated overnight at −20 °C. To purify the RNA, we used a EtOH precipitation method: samples were centrifuged (30 s at full speed at 4 °C), the supernatant was discarded and EtOH (1 mL at 80%) added. We repeated this step a second time with EtOH (1 mL at 70%). Finally, the supernatant was removed and the pellet, after a brief air dry (15–20 s) at RT, was resuspended in nuclease-free water. Libraries were prepared with the KAPA Stranded mRNASeq Library Preparation Kit (#KK8421) according to the manufacturer's protocol. Paired end sequencing was performed on an Illumina Hiseq4000 sequencer at the Genomic Technology Facility of the University of Lausanne. We obtained ~15–25 million PE reads per individual. Sequence reads have been deposited in NCBI Sequence Read Archive (SRA) under the accession number PRJNA823913.

We mapped RNA-seq reads on the softmask genome assembly using STAR v2.7 with the 2-pass mapping option and using the gene annotations to properly find exonic junctions[53]. Then, we used the program featureCounts from the Subread package to count the number of reads mapped on each exon for each gene[56]. Then using the count's files, we performed differential gene expression analyses using DESeq2[57] of sterile and infected workers collected at 2 and 11 h after treatment. *P* values of differential expression analyses were corrected for multiple testing with a false discover rate (FDR) of 5%.

## X-Ray Micro-CT imaging

To examine the internal morphology of the MG (Fig. 3a), we scanned and visualized an intermediate-sized worker (CASENT0744096), although multiple other workers of different sizes were scanned and visually inspected to confirm the morphology of the exemplar specimen was representative. The specimens were fixed in 90% ethanol, stained in 2 M iodine solution for a minimum of 7 days, then washed and sealed in a pipette tip with 99% ethanol for scanning. The scans were performed with the ZEISS Xradia 510 Versa 3D X-ray microscope

at the Okinawa Institute of Science and Technology Graduate University, Japan. We scanned the thorax (mesosoma) and whole body to visualize both the structure of the gland, and its location in the whole ant. The scan parameters for the thorax scan were: Voxel size, 4.404 µm; Exposure time, 6 s; Voltage, 40 kV; Power, 3 W; Source distance, 23.147 mm; Objective, 4x; Projections, 1601. For the full body scan, we used: Voxel size, 12.5 µm; Exposure time, 1 s; Voltage, 40 kV; Power, 3 W; Source distance, 25.03 mm; Objective, 0.4x; Projections, 1601. The 3D reconstruction was performed using the ZEISS Scout-and-Scan Control System Reconstructor software (ZEISS Microscopy, Jena, Germany). The MG structures were manually segmented using Amira (version 2019.2; Thermo Fisher Scientific, Berlin, Germany) and visualized using VGStudio3.4 (Volume Graphics GmbH, Heidelberg, Germany). The gland atrium was rendered using isosurface and a clipping plane was created to visualize the cross-section. All other materials were visualized using Phong volume rendering.

## Statistical analysis

For statistical analyses and graphical illustration, we used the statistical software R v4.1.0[58] with the user interface RStudio v1.4.1717 and the R package ggplot2 v3.3.5[59]. To select the appropriate statistical tests, we tested for deviations from the normal distribution with the Shapiro–Wilks test ($P > 0.05$). A Bartlett test was used to verify homoscedasticity ($P > 0.05$). All our models included colony as a random factor and an overall likelihood ratio test against an intercept only model. In case of multiple testing, a Holm-Bonferroni correction was performed with the adjusted P-values given throughout the text. To test for significant differences in the survival curves, we conducted mixed effect cox proportional hazards regression models (Supplementary Fig. 2) using the R package survminer (v0.4.9) followed by post-hoc analyses using least square means with the R package lsmeans (v.2.30; Supplementary Tables 3, 4 and 6). The survival curves were illustrated using Kaplan-Meier cumulative survival curves (Figs. 1c, 2a, and 4, and Supplementary Figs. 3 and 5). For the bacterial load (ΔCq) within the thorax content between treated and untreated infected individuals (Figs. 1a and 2b) we conducted linear mixed effect models with a least square means (lsmeans) post-hoc analysis. For the bacterial growth inhibition (Fig. 3d) we conducted a Mann–Whitney $U$ test. Differences in the CHC-profile composition were calculated using a permutational multivariate analysis of variance (ADONIS) on a Bray-Curtis dissimilarity matrix using the package vegan (v.2.5–7; Supplementary Table 7). For the different durations of MG care between sterile and infected individuals (Supplementary Fig. 4), we conducted a linear mixed effect model with individual as random factor followed by a Satterthwaite's t-test. For differences between chemical compound groups of the CHC profiles (Supplementary Fig. 6), we conducted an analysis of variance (AOV) followed by a Tukey Honest Significant differences test (Supplementary Table 9). For behavioral differences in wound care between sterile and infected individuals (Fig. 3b, c), we modeled wound care as a binary event using binomial generalized additive models with posthoc contrasts to identify intervals of time during which the probability of receiving wound care differed between sterile and infected individuals. See Supplementary Information for a reproducible modeling workflow.

## Ethics and inclusion statement

Our study brings together authors from a number of different countries but failed to include scientists based in the country where the study was carried out. We recognize that more could have been done to engage local scientists with our research as our project developed, and to embed our research within the national context and research priorities. Whenever possible, our research was discussed with local stakeholders to seek feedback on the questions to be tackled and the approach to be considered. Whenever relevant, literature published by scientists from the region was also cited; efforts were made to consider relevant work published in the local language.

## Reporting summary

Further information on research design is available in the Nature Portfolio Reporting Summary linked to this article.

## Data availability

The raw amplicon-sequence data generated in this study have been deposited at the Sequence Read Archive (SRA) under accession code PRJNA826317. The sequence reads data generated in this study have been deposited in NCBI Sequence Read Archive (SRA) under accession code PRJNA823913. The proteomics data generated in this study have been deposited at the ProteomeXchange Consortium via the PRIDE partner repository under accession code PXD033003. The CHC and MG data generated via GC-MS in this study have been deposited at the Dryad repository under the https://doi.org/10.5061/dryad.hqbzkh1j6 [https://datadryad.org/stash/share/gFLLhPMhmWW8HJ_JaKyTqg9Eq3 QXqGS_EgxYEQvKKe8][60]. The Genome Assembly data are available under restricted access due to it being part of another Publication within the GAGA Project[47], access can be obtained by contacting the corresponding Author of the GAGA Project. Source data are provided as a Source Data file. Source data are provided with this paper.

## Code availability

The R-code used in this study are provided in Supplementary data 1 and 2.

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

## Acknowledgements

We thank the Comoé National Park Research Station for the use of their facilities for the field and laboratory research and the park management of Office Ivoirien des Parcs et Réserves for facilitating field research in the park. We thank GAGA for providing the assembled *Megaponera analis* genome. We thank Barbara Milutinovic and the Cremer lab at IST Austria for teaching us the methodology for the antimicrobial assay.

We thank David Kouassi and Abou Ouattara for their help with nest excavation and colony maintenance in the field. We thank Christine La Mendola for her help with sample preparation for genetic analyses and Camille Lavoix for proof reading of the manuscript. This study was supported by the Swiss NSF grant 310030_156732 and the ERC Advanced Grant *resiliANT* (no: 741491) to LaK. ACL was supported by Swiss NSF grant PR00P3_179776. ETF was supported by the DFG Emmy Noether Programme (no: 511474012).

## Author contributions

Conceptualization: E.T.F., La.K. Methodology: E.T.F., Lu.K., J.L., Q.H., A.C.L., A.D., F.A., E.P.E., P.W., T.S., and D.B.S. Investigation: E.T.F., Lu.K., J.L., Q.H., A.C.L., E.P.E., and T.S. Visualization: E.T.F., Q.H., F.A., E.P.E., A.C.L., J.L., and D.B.S. Funding acquisition: La.K. and E.T.F. Project administration: E.T.F., P.E., and La.K. Supervision: E.T.F. and La.K. Writing – original draft: E.T.F. and La.K. Writing – review & editing: E.T.F., Lu.K., J.L., Q.H., A.C.L., A.D., F.A., E.P.E., P.W., P.E., T.S., D.B.S., and La.K.

## Funding

## Competing interests

The authors declare no competing interests.
