## [Peer Review File · Nature Communications]

Reviewers' Comments:

Reviewer #1:

Remarks to the Author:

An excellent paper, I have some clarifying comments below:

Line 69: Could you perhaps skip this part here, and go straight to the next section and say "To investigate if the differences in the bacterial data affect survival..." as it's confusing to look at the survival graph without the context of the rest of the experiment, as there are four lines, different nest treatments etc.

Line 74: Perhaps more information is needed here because the authors conclude that the workers are doing something here to the wounds, but this information is not given, so it leaves the reader wondering if the survival differences could be due to the nest environment or some other unmeasured factor. How similar were the nest and isolation setups? Did you have behavioural observations here? If not, then I think it's fine to reference your previous work that they groom the wounds, but something extra is needed here to make it a more convincing argument.

Line 96: No dotted lines in the figure but mentioned in the caption.

Line 123: Did you observe any other gland use behaviour or just MG? The venom gland often also has antimicrobial properties. Additionally, I wondered if you had a handling control for the plugging treatment (i.e. a random plug elsewhere on the body).

Line 124: Could say if the wounds were exposed to the pure bacteria or soil here.

Line 143: Is 33% in the control higher than you previously found? I.e. in Figure 2. Any reason why this might be?

Line 149: I think I would be more cautious here and call it a cue, since injury itself could trigger immunity pathways that affect CHCs (and you find differential immune gene expression). Moreover, you do not confirm these CHC changes trigger the behaviour, though this is likely, so I think calling it a cue would be more appropriate.

Line 225: In my opinion this closing statement is purely superficial and somewhat cheapens an otherwise excellent paper. I agree that there is a potential human application here and how ants deal with such issues without the evolution of antibiotic resistance is a great open question, but suggesting that these two processes are in anyway convergent is, I think, a mistake.

Line 320: What about the wound treaters? They're arguably the more important than the ones being injured; is there any evidence the minors are most often the wound groomers?

Line 335: is this true for the soil sample too?

Reviewer #2:

Remarks to the Author:

What are the noteworthy results?

While the first author had already shown that *Megaponera analis* ants care for their injured nestmates, tend to their wounds and that this wound care increases their survival rate (Frank et al. 2017 & 2018), this follow-up study further investigates how the ants fight bacterial infections in their nestmates. The authors identify the bacteria that infect injured ants and show how these bacteria threaten ant survival when they proliferate in infected ants. They also show experimentally that the secretion of the metapleural gland, which contains antimicrobial substances, effectively combats infection with pathogenic bacteria. They used proteomic and chemical analyses to investigate the composition of the secretion. However, they do not show which proteins and/or chemicals specifically are directly responsible for the antibiotic properties of the secretion of the metapleural gland. Finally, they use cuticular hydrocarbon analyses to

investigate whether infected workers chemically signal that they are injured or infected, which indeed seems to be the case. This information could be used by the ants' nest mates to take care of infected workers.

This study contains many novel findings. The results show that the ants show a level of care for infected group members that has rarely been shown outside of human societies. The antimicrobial and antifungal chemicals and proteins detected in the metapleural gland secretion might offer potential for further investigations of their potential usage in humans, which can also be infected by the most commonly found pathogenic soil bacterial, *Pseudomonas aeruginosa*.

Will the work be of significance to the field and related fields?

Yes, this paper will undoubtedly be widely cited and cause a stir beyond the field of social insect research.

How does it compare to the established literature?

As this study system is really unique, there are not so many publications on this topic. The only system investigated in that depth are interactions of ants with microorganisms is the analysis of leafcutter ants and their coevolution with their beneficial fungus. Here, the ants fight a parasitic fungus by using the secretion of the metapleural gland to support fungicide-producing bacteria. There are also a number of studies on the gut microbiome of ants and on the use of resin to keep ant colonies free of microbial pathogens. However, they are all not really comparable to this study.

Does the work support the conclusions and claims, or is additional evidence needed?

Yes this study contains many different analyses tackling all the relevant aspects. This is impressive work. Possibly in future it would be interesting to investigate which specific chemicals and proteins are responsible for the antibiotic effects and which genes underlay their production. Many proteins are uncharacterized so far, so there is a clear avenue for future findings. However, this paper clearly stands on its own.

Are there any flaws in the data analysis, interpretation and conclusions?

Do these prohibit publication or require revision?

Not really, seems competently done. The graphs could be improved a little or even aligned. For example, the description of the different treatments in Figure 1c and Figure 4 is different and less clear than in Figure 2a.

In Figure 1b the raw data points are shown in the boxplot, but unfortunately not in Figure 1a or Figure 3d. In addition, the labelling of Figure 2 a is not correct because, contrary to the legend, no dotted line is shown here.

Is the methodology sound? Does the work meet the expected standards in your field?

Yes

Is there enough detail provided in the methods for the work to be reproduced?

The methods are generally well described and are standard in the field.

The paper is a bit strangely written, as there is no introduction to the topic and the results are not properly discussed. It may be that this is because the manuscript is still in a Nature format, but this should be adjusted. A good introduction to the scientific topic and a proper discussion of the findings in contrast to the literature is really missing. Also the title could be more informative.

Reviewer #3:

Remarks to the Author:

In this manuscript, the authors studied the metapleural glands of the ant *Megaponera analis* using different approaches and methodologies. Overall, the experiments presented are very well done and are quite convincing.

I focused on the transcriptomic and proteomic aspects of the study. From a strictly technical point of view, these two approaches were remarkably well performed. Besides, the sequenced genome enabled to map the reads from transcriptomics and also to generate a protein database for proteomic analysis.

However, I have two main comments:

1) The bottom-up methodology used in proteomics is excellent if you want to identify large proteins, but it doesn't really allow you to identify small proteins such as peptides. Most

antimicrobial peptides described in literature (from hemolymph or venom) are polycationic. If such peptides are present in MG secretions, then the tryptic digestion used is not recommended for identifying this type of molecule. It would have been interesting to couple the analysis with a top-down approach where the proteins/peptides are not subjected to tryptic digestion.

2) The authors assume that MG secretions are different from those of fat bodies. Therefore, they use filtration to remove all proteins detected in both MG secretions and hemolymph. Isn't it possible that the same antimicrobial compounds secreted by fat bodies are also secreted by MGs?

Response to the Reviewers

Reviewer #1 (Remarks to the Author):

An excellent paper, I have some clarifying comments below:

1. Line 69: Could you perhaps skip this part here, and go straight to the next section and say "To investigate if the differences in the bacterial data affect survival..." as it's confusing to look at the survival graph without the context of the rest of the experiment, as there are four lines, different nest treatments etc.

We agree, and removed the sentence. It now reads:

Line 81: To investigate if the differences in bacterial abundance and composition affect survival, we either placed infected and sterile ants in their colony or kept them in isolation.

2. Line 74: Perhaps more information is needed here because the authors conclude that the workers are doing something here to the wounds, but this information is not given, so it leaves the reader wondering if the survival differences could be due to the nest environment or some other unmeasured factor. How similar were the nest and isolation setups? Did you have behavioural observations here? If not, then I think it's fine to reference your previous work that they groom the wounds, but something extra is needed here to make it a more convincing argument.

We agree that the wound care behavior was not properly described in the introduction. Following the suggestion of Reviewer #2 we added more information into the Introduction, explaining the process of wound care behavior in more details.

Concerning behavioral data of ants inside the nest for the survival assay of Fig. 1c: the wound care behavior was similar to what is described in Frank et al. 2018. We also observed the use of the Metapleural gland but we did not quantify the frequency. The behavioral data presented in Fig.3 were collected together with the data for the survival assay of Fig. 2a. We also added a sentence to the end of the paragraph to emphasize the importance of wound grooming on survival:

Line 90: Overall, these data demonstrate that *M. analis* workers are capable of effectively treating wounds that have been exposed to soil pathogens through social interactions.

3. Line 96: No dotted lines in the figure but mentioned in the caption.

Thank you for noticing, this was removed.

4. Line 123: Did you observe any other gland use behaviour or just MG? The venom gland often also has antimicrobial properties. Additionally, I wondered if you had a handling control for the plugging treatment (i.e. a random plug elsewhere on the body).

We never observed the venom gland (or other glands) to be used during wound care but we agree that it could potentially also be used given the antimicrobial properties.

Concerning the handling control, we put a color dot on the back of the thorax in the unplugged treatment.

5. Line 124: Could say if the wounds were exposed to the pure bacteria or soil here.

It was pure bacteria. We have changed the sentence to clarify this.

Line 115: To study the proximate mechanisms reducing mortality of ~~infected~~ ants infected with *P. aeruginosa* when they are returned to their nestmates, we introduced injured ants (with sterile and infected wounds) to their nestmates and filmed them for 24 hours.

6. Line 143: Is 33% in the control higher than you previously found? I.e. in Figure 2. Any reason why this might be?

While mortality in the control is higher in Fig.4 than Fig. 2, this difference is not significant. A possible explanation for the small differences could be variations in colony condition, or an effect of the season and year across the experiments. Data collection for Fig. 2 was in March 2019, while for Fig. 4 it was in October 2022.

7. Line 149: I think I would be more cautious here and call it a cue, since injury itself could trigger immunity pathways that affect CHCs (and you find differential immune gene expression). Moreover, you do not confirm these CHC changes trigger the behaviour, though this is likely, so I think calling it a cue would be more appropriate.

We agree and have changed the text as suggested.

Line 142: Because cuticular hydrocarbons (CHCs) are known to be frequently used as a source of information in ants¹¹, we investigated whether the ~~CHC profile~~ of infected ants ~~could signal their injured state through~~ changed ~~in the profile~~ over the course of the infection.

and

Line 157: Consistent with the idea that changes in CHC profiles could provide ~~information cues~~ on the health status of ants¹², the observed differences in the CHC profiles mostly stemmed from differences in the relative abundance of alkadienes (Extended Data Fig. 6, Supplemental Table 9), which are among the CHC compounds most relevant for communication in social insects¹³.

8. Line 225: In my opinion this closing statement is purely superficial and somewhat cheapens an otherwise excellent paper. I agree that there is a potential human application here and how ants deal with such issues without the evolution of antibiotic resistance is a great open question, but suggesting that these two processes are in anyway convergent is, I think, a mistake.

We agree with the Reviewer, and removed the sentence. The ending now reads:

Line 218: Remarkably, the primary pathogen in ant's wounds, *Pseudomonas aeruginosa*, is also a leading cause of infection in combat wounds, where infections can account for 45% of casualties in humans²⁴. ~~In *M. analis* the targeted treatment with antimicrobial compounds, which was previously thought to be unique to humans, was extremely effective in preventing lethal bacterial infections with *P. aeruginosa*. This could potentially lead to promising new medical compounds to cure infections in human societies. This demonstrates convergence in both the challenges of warfare and the solutions that evolved to mediate them in human and insect societies.~~

9. Line 320: What about the wound treaters? They're arguably more important than the ones being injured; is there any evidence the minors are most often the wound groomers?

This is an excellent question for future studies. Unfortunately, we were not able to follow individually the caretakers in the present study. From our observations, it does seem that workers of all sizes were participating in the wound care behavior and that there was no size class preferentially being involved.

10. Line 335: is this true for the soil sample too?

Yes. We clarified this in the text:

Line 247: To quantify the lethality of various types of pathogens, workers were injured by a sterile cut in the middle of the femur on the hind leg and had the fresh wound submerged for 2 seconds in a 10 μ L phosphate buffered saline (PBS) solution with a known pathogen concentration (approx. 10^6 bacteria of either the mix of soil pathogens or isolated pathogens).

Reviewer #2 (Remarks to the Author):

11. What are the noteworthy results?

While the first author had already shown that *Megaponera analis* ants care for their injured nestmates, tend to their wounds and that this wound care increases their survival rate (Frank et al. 2017 & 2018), this follow-up study further investigates how the ants fight bacterial infections in their nestmates. The authors identify the bacteria that infect injured ants and show how these bacteria threaten ant survival when they proliferate in infected ants. They also show experimentally that the secretion of the metapleural gland, which contains antimicrobial substances, effectively combats infection with pathogenic bacteria. They used proteomic and chemical analyses to investigate the composition of the secretion. However, they do not show which proteins and/or chemicals specifically are directly responsible for the antibiotic properties of the secretion of the metapleural gland. Finally, they use cuticular hydrocarbon analyses to investigate whether infected workers chemically signal that they are injured or infected, which indeed seems to be the case. This information could be used by the ants' nestmates to take care of infected workers. This study contains many novel findings. The results show that the ants show a level of care for infected group members that has rarely been shown outside of human societies. The antimicrobial and antifungal chemicals and proteins detected in the metapleural gland secretion might offer potential for further investigations of their potential usage in humans, which can also be infected by the most commonly found pathogenic soil bacterial, *Pseudomonas aeruginosa*.

We are pleased you enjoyed our study. We completely agree that further studies into the individual chemical compounds and proteins are necessary to better understand their antimicrobial properties but believe that is beyond the scope of this study.

12. Will the work be of significance to the field and related fields?

Yes, this paper will undoubtedly be widely cited and cause a stir beyond the field of social insect research.

Thank you for sharing our view concerning the impact of this study.

13. How does it compare to the established literature?

As this study system is really unique, there are not so many publications on this topic. The only system investigated in that depth are interactions of ants with microorganisms is the analysis of leafcutter ants and their coevolution with their beneficial fungus. Here, the ants fight a parasitic fungus by using the secretion of the metapleural gland to support fungicide-producing bacteria. There are also a number of studies on the gut microbiome of ants and on the use of resin to keep ant colonies free of microbial pathogens. However, they are all not really comparable to this study.

We agree and therefore added a few sentences in the discussion to address these other observations (see our answer to Comment #18 by Reviewer #2).

14. Does the work support the conclusions and claims, or is additional evidence needed?

Yes this study contains many different analyses tackling all the relevant aspects. This is impressive work. Possibly in future it would be interesting to investigate which specific chemicals and proteins are responsible for the antibiotic effects and which genes underlay their production. Many proteins are uncharacterized so far, so there is a clear avenue for future findings. However, this paper clearly stands on its own.

We agree and hope that future studies will address these issues.

15. Are there any flaws in the data analysis, interpretation and conclusions? Do these prohibit publication or require revision?

Not really, seems competently done. The graphs could be improved a little or even aligned. For example, the description of the different treatments in Figure 1c and Figure 4 is different and less clear than in Figure 2a.

In Figure 1b the raw data points are shown in the boxplot, but unfortunately not in Figure 1a or Figure 3d. In addition, the labelling of Figure 2 a is not correct because, contrary to the legend, no dotted line is shown here.

Thank you for mentioning these inconsistencies. We have improved the figure legends for readability and added the individual data points for figure 1a, 2b and 3d. We also removed the mention of the dotted line in the legend of Figure 2.

The figure legends now read:

Fig. 1 | Lethal effects and diversity of soil pathogens. (a) Relative 16S rRNA gene copies (bacterial load ΔCq) for individuals whose wounds were exposed to a sterile PBS solution (Sterile), or soil pathogens diluted in PBS (Infected, OD=0.1), 2 and 11 hours after exposure (see Supplementary Table 1 for statistical results). $n=10$ per boxplot. Significant differences ($P<0.05$) are shown with different letters. **(b)** Absolute 16S rRNA gene copy numbers summarized at the genus-level for the 18 amplicon sequence variants (ASV) that had at least 1% relative abundance in five of the 40 analyzed ants across the experiment ($n=10$ per boxplot). Significance is shown for pairwise comparisons between sterile and infected ants: ***= $P<0.001$; **= $P<0.01$; *= $P<0.05$; n.s.= not significant ($P>0.05$). Detailed statistical results in Supplementary Table 2 **(c)** Kaplan – Meier cumulative survival rates of workers in isolation (dotted line) or inside the nest (solid line) whose wounds were exposed to a sterile PBS solution (Sterile), or soil pathogens diluted in PBS (Infected, OD=0.1) ~~the same way as in Fig. 1a (infected or sterile)~~. Significant differences ($P<0.05$) are indicated with different letters (detailed statistical results in Supplementary Fig. 2a and Supplementary Table 3).

Fig. 2 | Survival probability and pathogen load of sterile and infected ants. (a) Kaplan – Meier cumulative survival rates of workers in isolation ~~(dotted line)~~ or inside the nest ~~(solid line)~~ whose

wounds were exposed to *P. aeruginosa* diluted in PBS (Infected, OD=0.05) or a sterile PBS solution (Sterile). Detailed statistical results in Supplementary Fig. 2c and Supplementary Table 3. **(b)** relative bacterial load (ΔCq) of *Pseudomonas* at two different time points (2h and 11h) for ants in isolation or inside the nest with wounds treated the same way as in Fig. 2a (Infected or Sterile). $n=6$ per boxplot, significant differences ($P<0.05$) are shown with different letters (Supplementary Table 4).

Fig. 4 | Effect of MG secretions on survival of sterile and infected ants. Kaplan – Meier cumulative survival rates of workers inside sub-colonies with a plugged MG opening (dotted line) or with an unmanipulated MG opening (solid line) whose wounds were exposed to a sterile PBS solution (Sterile $n=12$) or *P. aeruginosa* diluted in PBS (OD=0.05) in PBS (Infected $n=12$). Detailed statistical results in Supplementary Fig. 2d and Supplementary Table 5. Additional data for ants in isolation can be found in Supplementary Fig 5.

16. Is the methodology sound? Does the work meet the expected standards in your field?
Yes

Thank you.

17. Is there enough detail provided in the methods for the work to be reproduced?
The methods are generally well described and are standard in the field.

Thank you.

18. The paper is a bit strangely written, as there is no introduction to the topic and the results are not properly discussed. It may be that this is because the manuscript is still in a Nature format, but this should be adjusted. A good introduction to the scientific topic and a proper discussion of the findings in contrast to the literature is really missing. Also the title could be more informative.

We agree and added information into the introduction and discussion. We have also changed the title, which now reads: Antimicrobial wound care of infected injuries in an ant society.

Added sections to the Introduction:

Line 31: Infections are a major mortality risk in animals^{1,2}, with the risk of transmission of contagious pathogens being particularly life-threatening in group living animals³. This has led to a suite of pathogen-induced changes in social interactions, like social distancing, sickness cues and medical care⁴⁻⁶. In injured individuals, the major barrier against infections (the cuticle or epidermis) is damaged and therefore provides an easy entry point for life-threatening infections⁷. ~~Infected wounds are a major mortality risk for animals^{1,2}, but the identification and medicinal treatment of infected wounds has thus far been considered a uniquely human behavior.~~ Recently, several mammals have been shown to lick wounds to apply antiseptic saliva^{1,5}. However, the efficacy of these behaviors remains largely unknown and occur irrespective of the state of the wound.

In social insects, interactions to combat pathogens range from preventive measures like nest disinfection or allogrooming to moribund individuals leaving the nest to die in isolation or the destructive disinfection of their infected brood^{3,8-10}. But if and how social insect colonies care for injured individuals that were exposed to pathogens remains poorly understood. Workers of the predatory ant *Megaponera analis* have been shown to care for the injuries of nestmates^{2,7}, which are common because this ant feeds exclusively on pugnacious termite species¹¹. As many

as 22% of the foragers engaging in raids attacking termites have one or two missing legs². Injured workers are carried back to the nest where other workers treat their wounds, **by licking and grooming the wound during the first three hours after injury**⁷. When the wounds of injured workers are not treated by nestmates, 90% of the injured workers die within 24 hours after injury⁷, but the mechanisms behind these treatments are unknown. **The aim of this study is therefore to identify the cause of death in injured individuals and the potential mechanisms involved in the detection and treatment of injuries.**

Here we show that infections by the gram-negative bacterium *Pseudomonas aeruginosa* is the main cause of death in injured *M. analis* workers. We found that infected wounds are treated more often than sterile wounds and that this correlates with changes in the cuticular hydrocarbon profile of infected individuals. We describe the use of the metapleural gland to treat infected wounds and quantify its content, identifying 112 chemical compounds and 41 proteins in the gland's secretion, half of which have antimicrobial or wound healing properties. Overall, this study demonstrates that the targeted use of antimicrobials to treat infected wounds, previously thought to be a uniquely human behavior, has evolved in insect societies as well.

Added sections to the Discussion:

Line 197: The number of chemical compounds identified in the MG's secretions of *M. analis* (112) is far greater than in other ant species, where the number of compounds ranges between 1 and 35 and mostly consists of carboxylic acids rather than alkaloids or antibiotic-like compounds¹². **In other ant species, the compounds in MG secretions are generally believed to be effective against early developmental stages of pathogens** ~~While MG secretions have been observed to be used to,~~ like during sterilization of the nest or as a response to recent fungal exposure¹²⁻¹⁴ ~~in other ant species.~~ While we also observed a prophylactic use of MG secretions directly after injury, the most intense use of the MG was at the height of the infection (10-12 hours, Fig. 3c), a period where, without care, infected ants start to die (Fig. 2a). This suggests that, in addition to prophylactic substances, the MG secretions also contain antimicrobial compounds capable of combatting festering infections. ~~it had never been observed in the context of wound care.~~

During wound care, the use of the MG secretions probably fulfils a similar role as the mammals' antiseptic saliva. This analogous role likely led to the convergent evolution of functionally similar antimicrobial and wound healing proteins²⁷. However, wound care treatments are indiscriminate in mammals as they have never been observed to depend on the infected state of the wound.

Conclusion

This study reveals a highly effective behavioral adaptation to identify and treat festering infections of open wounds in a social insect. The prophylactic and therapeutic use of antimicrobial secretions to counteract infection in *M. analis* (Fig. 3c) mirrors modern medical procedures used for treating dirty wounds²⁸. Remarkably, the primary pathogen in ant's wounds, *Pseudomonas aeruginosa*, is also a leading cause of infection in combat wounds, where infections can account for 45% of casualties in humans²⁹. **In *M. analis* the targeted treatment with antimicrobial compounds, which was previously thought to be unique to humans, was extremely effective in preventing lethal bacterial infections by *P. aeruginosa*. This could potentially lead to promising new medical compounds to cure infections in human societies.** ~~This demonstrates convergence in both the challenges of warfare and the solutions that evolved to mediate them in human and insect societies.~~

Reviewer #3 (Remarks to the Author):

In this manuscript, the authors studied the metapleural glands of the ant *Megaponera analis* using different approaches and methodologies. Overall, the experiments presented are very well done and are quite convincing.

I focused on the transcriptomic and proteomic aspects of the study. From a strictly technical point of view, these two approaches were remarkably well performed. Besides, the sequenced genome enabled to map the reads from transcriptomics and also to generate a protein database for proteomic analysis.

Thank you for these kind comments.

However, I have two main comments:

19. The bottom-up methodology used in proteomics is excellent if you want to identify large proteins, but it doesn't really allow you to identify small proteins such as peptides. Most antimicrobial peptides described in literature (from hemolymph or venom) are polycationic. If such peptides are present in MG secretions, then the tryptic digestion used is not recommended for identifying this type of molecule. It would have been interesting to couple the analysis with a top-down approach where the proteins/peptides are not subjected to tryptic digestion.

This is an excellent point, and we agree that further exploration is warranted. However, we believe that it is beyond the scope of the present study. The reason for the use of our method was that it was difficult to collect the secretions directly. Consequently, we took the triangulation approach to see what was common across the wash, the glandular tissue and only minimally present in the hemolymph.

20. The authors assume that MG secretions are different from those of fat bodies. Therefore, they use filtration to remove all proteins detected in both MG secretions and hemolymph. Isn't it possible that the same antimicrobial compounds secreted by fat bodies are also secreted by MGs?

We agree that it is possible that antimicrobial compounds found in the hemolymph are also present in the MG. We decided to use a conservative method by looking at what is unique in the MG, due to the difficulties of extracting the MG secretions without contaminations.

Reviewers' Comments:

Reviewer #1:

Remarks to the Author:

The authors have fully addressed my comments and I recommend publication.

Reviewer #2:

Remarks to the Author:

The authors revised the manuscript carefully following my previous comments. In particular, the title is now more informative and there is a proper introduction and a short discussion.

I have only one minor issue. At the end of the introduction, the authors provide a short summary of the main finding and write: „Here we show that infections by the gram-negative bacterium *Pseudomonas aeruginosa* is the main cause of death in injured *M. analis* workers.“

When I read this sentence, I sounds as if the authors have field data for this. What I understand however, ist hat they infected injured workers by using the soil, which contain *Pseudomonas aeruginosa* and which did infect the ants then. In the field, this might be also a common infection and cause of death, but we do not know this. I would suggest that the authors are more specific here.

Reviewer #3:

Remarks to the Author:

The authors have answered my questions and I have no further comments on this new version.

This manuscript will contribute greatly to our knowledge of metapleuristic glands. I therefore recommend its publication.

Response to the Reviewers

Reviewer #1 (Remarks to the Author):

The authors have fully addressed my comments and I recommend publication.

Thank you.

Reviewer #2 (Remarks to the Author):

The authors revised the manuscript carefully following my previous comments. In particular, the title is now more informative and there is a proper introduction and a short discussion.

Thank you. We further adapted the title with the guidance of the Editor, it now reads:
Targeted treatment of injured nestmates with antimicrobial compounds in an ant society

I have only one minor issue. At the end of the introduction, the authors provide a short summary of the main finding and write: „Here we show that infections by the gram-negative bacterium *Pseudomonas aeruginosa* is the main cause of death in injured *M. analis* workers.“

When I read this sentence, I sounds as if the authors have field data for this. What I understand however, is that they infected injured workers by using the soil, which contain *Pseudomonas aeruginosa* and which did infect the ants then. In the field, this might be also a common infection and cause of death, but we do not know this. I would suggest that the authors are more specific here.

We agree, we changed it to read:

Line 56: Here we show that the gram-negative bacterium *Pseudomonas aeruginosa* ~~is the main cause of death~~ can cause lethal infections in injured *M. analis* workers.

Reviewer #3 (Remarks to the Author):

The authors have answered my questions and I have no further comments on this new version. This manuscript will contribute greatly to our knowledge of metapleuristic glands. I therefore recommend its publication.

Thank you.